# Transport and Electrochemical Characteristics of CJMCED Homogeneous Cation Exchange Membranes in Sodium Chloride, Calcium Chloride, and Sodium Sulfate Solutions

**DOI:** 10.3390/membranes10080165

**Published:** 2020-07-25

**Authors:** Veronika Sarapulova, Natalia Pismenskaya, Dmitrii Butylskii, Valentina Titorova, Yaoming Wang, Tongwen Xu, Yang Zhang, Victor Nikonenko

**Affiliations:** 1Department of Physical Chemistry, Kuban State University, 149 Stavropolskaya st., 350040 Krasnodar, Russia; vsarapulova@gmail.com (V.S.); dmitrybutylsky@mail.ru (D.B.); valentina.titorova@mail.ru (V.T.); v_nikonenko@mail.ru (V.N.); 2CAS Key Laboratory of Soft Matter Chemistry, Collaborative Innovation Center of Chemistry for Energy Materials, School of Chemistry and Material Science, University of Science and Technology of China, Hefei 230026, China; ymwong@ustc.edu.cn (Y.W.); twxu@ustc.edu.cn (T.X.); 3School of Environmental and Safety Engineering, Qingdao University of Science and Technology, 53 Zhenzhou Road, Qingdao 266042, China; zhangyang@qust.edu.cn

**Keywords:** cation exchange membrane, electric conductivity, diffusion permeability, selectivity, current–voltage characteristic

## Abstract

Recently developed and produced by Hefei Chemjoy Polymer Material Co. Ltd., homogeneous CJMC-3 and CJMC-5 cation-exchange membranes (CJMCED) are characterized. The membrane conductivity in NaCl, Na_2_SO_4_, and CaCl_2_ solutions, permeability in respect to the NaCl and CaCl_2_ diffusion, transport numbers, current–voltage curves (CVC), and the difference in the pH (ΔpH) of the NaCl solution at the desalination compartment output and input are examined for these membranes in comparison with a well-studied commercial Neosepta CMX cation-exchange membrane produced by Astom Corporation, Japan. It is found that the conductivity, CVC (at relatively low voltages), and water splitting rate (characterized by ΔpH) for both CJMCED membranes are rather close to these characteristics for the CMX membrane. However, the diffusion permeability of the CJMCED membranes is significantly higher than that of the CMX membrane. This is due to the essentially more porous structure of the CJMCED membranes; the latter reduces the counterion permselectivity of these membranes, while allowing much easier transport of large ions, such as anthocyanins present in natural dyes of fruit and berry juices. The new membranes are promising for use in electrodialysis demineralization of brackish water and natural food solutions.

## 1. Introduction

Recently, ion-exchange membranes (IEMs) have been widely used in capacitive deionization [1], electrolysis [2], Donnan [3] and neutralization dialysis [4], fuel cells [5,6], and bioelectrochemical systems [7]. They are utilized for extracting valuable components, such as ammonia, along with producing electricity [8], and in other applications. At the same time, electrodialysis (ED) is a traditional area of their application. In this process, under the action of an electric field applied by two electrodes, cations are removed from the feed solution through cation-exchange membranes (CEMs) permeable nearly exclusively to cations, and the anions are removed through anion-exchange membranes (AEMs) permeable nearly exclusively to anions. One of the first applications of ED (in 1902) was the purification of sugar syrup from mineral impurities [9]. Currently, ED is used to produce table salt [10], denitrify drinking water [11], isolate nutrients and organic compounds from their salt solutions [12,13,14], and to produce high-quality drinking water and water for irrigation in areas with insufficient rainfall [15]. ED is being actively introduced into the dairy industry [16], winemaking [17], juice conditioning industry [18], soybean processing [19], sauce production [20], and the extraction of dietary supplements and valuable medicines from agricultural waste [21,22]. This method allows the concentration of reverse osmosis (RO) retentates up to 150–300 g dm^−3^, which makes it possible to significantly reduce discharge of moderately concentrated (up to 30 g dm^−3^) effluents into the environment and helps to reduce the cost and improve the environmental feasibility of hybrid RO+ED processes [23]. Another important practical problem that many laboratories are trying to solve is the development of hybrid membrane technologies for the extraction of lithium from seawater [24] and salt lakes [25], as well as from industrial solutions [26].

Lately, interest in such processes has grown immensely due to the rapid development of selective ED and ED metathesis [27,28,29], which allows solving the most difficult problem of sedimentation, the danger of which is very great when processing multicomponent solutions. At the first stage of such a process, singly charged ions are selectively extracted from the mixed solution [30]. Then, the solution not containing sparingly soluble salts may undergo further separation and/or strong concentration.

It is known that the contribution of the cost of IEMs to the cost of the final product obtained in the process of ED can reach 40–50% [31]. This circumstance, as well as the continuously expanding fields of ED and other processes application, induced a rapid increase in the number of works on the development of new IEMs. Reviews of these works can be found in [32,33,34]. One of the recent developments is the CEMs manufactured by Hefei Chemjoy Polymer Material Co. Ltd. These CEMs have already been successfully used in conventional ED to extract gamma-aminobutyric acid [35], methylsulfonylmethane [36], sugar, sweet ingredients, and animal and plant extracts [37] from reaction mixtures that contain mineral impurities, as well as in reverse electrodialysis used as a renewable energy supply when processing mixed solutions of chloride with sulfate and humic acid [38]. These membranes have been also applied to salt concentration [39], separation in bioproducts [40,41,42], and desalination of plant extract [43].

However, despite many examples of their application, knowledge of the structure, transport, and electrochemical properties of these membranes is far from complete.

In this work, we report the results of a study of the transport characteristics (electric conductivity, diffusion permeability, transport numbers) and current–voltage curves of cation-exchange membranes CJMC-3 and CJMC-5 manufactured by Hefei Chemjoy Polymer Material Co. Ltd., which could help to determine the possible areas of their application.

Generally, CEMs (as well as the AEMs) are divided into homogeneous and heterogeneous types, according to their structure and the way of fabrication. Homogeneous membranes are conventionally produced by the “paste” method. They contain a homogeneous at the nanoscale level (up to 100 nm) single-phase ion-exchange matrix, which can be a single solid electrolyte (such Nafion^TM^, DuPont Co., Wilmington, CA, USA; MF-4SK, JSC NPO Plastpolymer, Saint Petersburg, Russia) or include reinforcing fabric (some kinds of Nafion^TM^ membranes (Nafion™ N-324, Nafion™ N424 and others), Neosepta AMX, CMX, Astom, Tokyo, Japan; CJMCED and CJMAED, Hefei ChemJoy Polymer Materials, Hefei, China; and others). Heterogeneous IEMs consist of micrometer-sized (from 5 to 50 microns) ion-exchange polymer particles incorporated into an inert binder (MK-40, MA-40, Shchekinoazot, Russia; Ralex MH-PES, MEGA a.s., Czech Republic; FTAM-E, FTAM-A, FuMA-Tech GmbH, Ludwigsburg, Germany) [44]. In homogeneous IEMs, the functional charged groups are chemically bonded to the matrix, which makes a single phase extend throughout the entire membrane; in heterogeneous membranes, the charged groups are chemically bonded to the matrix; however, the size is limited by the size of the resin particles, and different particles are physically mixed with the inert binder [45]. Therefore, in homogeneous membranes, one polymer fulfils two functions: granting the ion-exchange capability and the structural support; in heterogeneous membranes, these two functions are divided between two different polymers [46]. Competition between homogeneous and heterogeneous IEMs lasts for decades. Heterogeneous membranes are always thicker and consume more energy. Their greater pores allow lower counterion permselectivity. The low conductive surface area fraction of these membranes causes higher concentration polarization and hence greater voltage and water splitting rate at the same average current density [47]. On the other hand, heterogeneous IEMs are less costly and sometimes more chemically stable.

The aim of this study is the assessment of some properties of the recent CJMCED membranes and comparison of them with those of a well-established commercial Neosepta CMX membrane. Although the CJMCED membranes are related to the same class as the CMX membrane (homogeneous with reinforcing fabric), the cost of CJMCED membranes is close to that of heterogeneous membranes. Besides, these membranes are more porous, which has its advantages and disadvantages.

## 2. Materials and Methods

### 2.1. Membranes

The CEMs under study are listed in Table 1. The homogeneous CJMC-3 and CJMC-5 membranes are manufactured by Hefei Chemjoy Polymer Materials Co. Ltd. (Hefei, China). These membranes are produced by the casting method [35,48]. They are reinforced with polyester fabric by hot rolling. The ion-exchange matrix of these membranes contains polyvinylidene fluoride (PVDF) functionalized with sulfonic groups, –SO_3_^–^ [35,49]. The side chains of CJMC-5 matrix are self-crosslinking with cross-linking agent sodium 4-styrenesulfonate (SSS) [50]. 

For comparison, we also present the characteristics of a commercial Neosepta CMX cation-exchange membrane produced by the ASTOM Corporation (Tokyo, Japan), which is widely used for potassium nitrate synthesis by electrodialysis-metathesis [28], in desalination and reverse electrodialysis [33], separation of methionine using bipolar membrane electrodialysis [40], electrodialysis for the separation of 5′-ribonucleotides from hydrolysate [49], and other applications. It is one of the best high-performance CEMs on the market. Like the novel CJMCED membranes, the CMX membrane is classified as homogeneous; it has functional sulfonic groups and is fabric-reinforced. This membrane is made using the “paste method” [51]. The ion-exchange composite material of this membrane consists of two interpenetrating phases [52]: a sulfonated styrene–divinylbenzene cross-linked copolymer (ion-exchange material) and polyvinyl chloride, PVC, whose particle diameter does not exceed 60 nm. The reinforcing PVC fabric is introduced into the membrane at the stage of polymerization of the ion-exchange matrix. 

### 2.2. Reagents

In experiments, we use: distilled water (electric conductivity of 1.1 ± 0.1 μS cm^−1^; pH = 5.5; 25 °C), solid NaCl and Na_2_SO_4_ of the analytical grade, as well as chemically pure solid CaCl_2_ (Vecton JSC, St. Petersburg, Russia). Solutions of these salts had a pH 5.4 ± 0.3 (NaCl) and 5.6 ± 0.3 (Na_2_SO_4_). The pH of the CaCl_2_ solution increased from 6.3 (0.02 mol dm^−3^) to 9.0 (1.0 mol dm^−3^), respectively. The diffusion coefficients of these electrolytes at infinite dilution are equal to, cm^2^ s^−1^ [55]: 1.61 × 10^−5^ (NaCl); 1.34 × 10^−5^ (CaCl_2_); 1.23 × 10^−5^ (Na_2_SO_4_). The values of crystallographic radii, Stokes radii, hydration energies and numbers, as well as diffusion coefficients of the ions that compose NaCl, CaCl_2_, and Na_2_SO_4_ are given in Appendix A. The electrolytes are chosen because their ions are most often present in natural and waste waters. 

### 2.3. Methods of Membrane Characterization

The standard salt pretreatment of membranes in NaCl solutions was carried out before the experiments [56]. The thickness and the total exchange capacity ($Q_{sw}$) of swollen membranes and the water content (*W*) were found using standard methods [57], the details of which are given in Appendix A. The distilled water contact angles were measured using the method of a resting drop [58]. The SOPTOP CX40M optical microscope (Yuyao, Zhejiang, P.R. China) with a digital eyepiece USB camera (5×, 10×, 20×, and 50× magnification) was used for visualization of the surface and cross-section of swollen membranes. The differential method with a clip cell [59,60] was applied to determine concentration dependences of the membrane electrical conductivity ($\kappa^{*}$). The measurements were made with an immittance meter AKIP 6104 (B+K Precision Taiwan, Inc., New Taipei City, Taiwan) at an alternating current frequency of 1 kHz.

Processing these dependences using the microheterogeneous model [61] allows determining the volume fractions of the gel phase ($f_{1}$) and the phase of the electroneutral solution filling the intergel spaces ($f_{1}$) in the studied membranes, as well as the electrical conductivity ($\bar{\kappa }$), the ion-exchange capacity of the gel phase ($\bar{Q}$) of these membranes, and the diffusion coefficients of the counterions Na^+^, Ca^2+^ ($\bar{D_{i}}$) in the gel phase. The microheterogeneous model considers the membrane as a two-phase system with volume fractions *f*_1_ and *f*_2_ of the corresponding phases $f_{1}\;$+$\;f_{2}$ = 1. The gel phase is a microporous swollen medium that consists of a polymer matrix with fixed groups, whose charge is counteracted by the charged solution containing mobile counterions and, to a lesser extent, coions. The threads of the reinforcing cloth and the inert filler (if any) are also included in the gel phase. The intergel spaces (the central parts of the meso- and macropores, including structural defects of the membrane) are filled with an electrically neutral solution, which is considered identical to the external solution. The details of the application of the microheterogeneous model for determining the structural-kinetic characteristics of membranes are described in many works, for example, in [62]. In this paper, we provide them in Appendix A.

A two-compartment flow cell [62] was used for determining the integral diffusion permeability coefficient of membranes (P), which is usually defined as [56]:(1)$$P=j\frac{d}{C}$$
where j is the flux density of an electrolyte measured when it diffuses through the membrane from the compartment containing a solution of concentration C into the compartment with initially distilled water; d is the membrane thickness. The confidence interval of the measurements of P is equal to 0.4 × 10^−8^ cm^2^ s^−1^. The scheme of the cell, the details of the measurements, and data processing are given in Appendix A. P is a characteristic convenient for the practical use. However, in theoretical considerations, the differential (or local) diffusion permeability coefficient, $P^{*}$, is often applied. The relation between these characteristics is given as $P=\frac{1}{C}\int_{0}^{C}P^{*}dC$ [56,61], which leads to the following formula [61]: (2)$$P^{*}=P+C\frac{dP}{dC}$$

In practice, for calculation $P^{*}$, it is more convenient to use the relationship between $P^{*}$ and P in the form [63]: (3)$$P^{*}=P\left(\beta +1\right)$$
with β=dlgP/dlgC, which is the slope of the lgP vs. lgC dependence. 

Knowing the values of $\kappa^{*}$ and $P^{*}$, it is possible to find the transport numbers of counterions ($t_{1}^{*}$) and coions ($t_{A}^{*}$) in a membrane [64]:(4)$$t_{1}^{*}=\frac{1}{2}+\sqrt{\frac{1}{4}-\frac{\left(z_{1}\left|z_{A}\right|\right)P^{*}F^{2}C}{\left(z_{1}+\left|z_{A}\right|\right)RT\kappa^{*}}}\;t_{A}^{*}=1-t_{1}^{*}$$
where $z_{1}$ and $z_{A}$ are the charge numbers of the counterion and coion, respectively, F is the Faraday constant, R is the gas constant, and T is the temperature.

The galvanodynamic current–voltage characteristics (CVC) of the membranes were obtained using a laboratory four-compartment flow-through cell [65]) shown in Figure 1. The current density was applied with the Autolab PGStat-100 (Metrohm Autolab B.V., Kanaalweg, The Netherlands) electrochemical complex, which was used also for recording the potential difference between the Ag/AgCl electrodes with the Luggin capillaries shown in Figure 1. The capillary tips were installed at a distance of about 0.8 mm from each side of the CEM under study (marked as CEM* in Figure 1). In addition to CVC, we have measured the change of pH of the feed solution, ΔpH, caused by its passage through the desalination channel (DC) of the cell. For this, pH measurements were carried out in two special flow-through cells equipped with a combination electrode; one of them was installed at the input, and the other at the output of the DC. Two auxiliary heterogeneous membranes (a MK-40 cation-exchange one and a MA-41 anion-exchange one, manufactured by Shchekinoazot JSC, Pervomaysky, Russia) were installed between the CEM under study. This allowed avoiding the transfer of the products of electrode reactions (the H^+^ and OH^−^ ions) from the electrode compartments to the compartments next to CEM*. Thus, the ΔpH value was due only to water splitting reactions occurring on the membranes forming the DC. The complete setup for electrochemical measurements is shown in Appendix A. 

Although the paper is devoted to the study of CEMs, the properties of the MA-41 can be of interest. This membrane is produced by hot rolling of powdered AV-17 anion-exchange resin with low-pressure polyethylene used as an inert binder [44]. The reinforcing nylon fabric is made of filaments with the diameter of 30–50 μm. The main characteristics of the MA-41 membrane are as follows: the thickness of wet membrane is 450 ± 50 μm; water content is equal to 24; IEC is equal to 1.22 ± 0.6 mmol g_sw_^−1^; counterion transport number is equal to 0.995 (at a 0.15 mol dm^−3^ NaCl solution).

The intermembrane distance, h, was 6.5 mm; the average flow rate of the electrolyte solution, V, was 0.4 cm s^−1^; the area of the polarized portion of the membrane was 2 × 2 cm^2^; Luggin capillaries (2) were located at a distance of about 0.8 mm from the membrane surface.

The reduced potential drop $\Delta \varphi^{\prime }$ [66]:(5)$$\Delta \varphi^{\prime }=\Delta \varphi -iR_{ef}$$
was used instead of the total potential drop Δφ, to exclude from consideration the system resistance at low currents, $R_{ef}$ (Ohm·cm^2^), which depends on the distance between the membrane and Luggin capillaries, membrane thickness, diffusion resistance of interphase boundaries, and other parameters that are hard to take into account when moving from one membrane system to another. The value of $R_{ef}$ was found by extrapolating the initial portion of the CVC (i→*0*) in coordinates i vs. dφ/di where i is the current density.

The value of the theoretical limiting current density was calculated according to the Lévêque equation obtained in the framework of the convective-diffusion model [67]:(6)$$i_{lim}^{Lev}=1.47\frac{FDz_{1}C_{1}}{h\left(1-t_{1}\right)}{\left(\frac{h^{2}V}{LD}\right)}^{1/3}$$
where F is the Faraday constant; D is the electrolyte diffusion coefficient; $z_{1}$ is the charge number of the counterion; $C_{1}$ is the molar concentration of the counterion in the solution at the entrance to the desalination compartment; h is the intermembrane distance; $t_{1}$ is the electromigration transport number of the counterion in the solution; V is the average linear flow velocity of the solution; L is the length of the membrane working area.

## 3. Results

### 3.1. Structural Characteristics of the Investigated Membranes

Figure 2 shows optical images of the surface and cross-sections of the CJMCED membranes. Both membranes are reinforced with the same reinforcing material (polyester), whose cell pitch equals 130 μm (Figure 2a,c). In the thinner (Table 1) CJMC-5 membrane, the reinforcing cloth fills the entire cross-section of the homogeneous ion-exchange material (Figure 2d). In the thicker (Table 1) CJMC-3 membrane, this cloth is shifted towards one of the surfaces (surface I) of the membrane (Figure 2b). The other surface (surface II) is smoother (Figure 2a). Intersections of the cloth threads are close to the surfaces in both membranes, making the surface undulated. The distance, *b*, between the levels corresponding to the top of the “hills” and the bottom of the “valleys” (Figure 2b) of the CJMC-3 surface is about 35 μm, which is comparable with the value of this parameter for the CMX membrane, *b* = 45 ± 10 μm [68]. For the CJMC-5 membrane, *b* is much lower (Figure 2d).

It appears that extended (lengthy) macropores are formed between the reinforcing threads and the ion-exchange material of the CJMC-3 and CJMC-5 membranes. These pores can be visualized by video recording the drying of swollen samples in air using the optical microscope described in Section 2.3. Some of the frames of this video are presented in Figure 3a–d. In the case of swollen samples equilibrated with a 0.02 M sodium chloride solution (Figure 3a,d), these pores cannot be seen because they are filled with the solution, which is optically nontransparent in transmitted light. White bands appearing around the filaments of reinforcing fabric (Figure 3b,c,e,f) are caused by the air penetrating in the pores after water evaporation. The material of the filaments is seen as the dark regions, which are cleared by a layer of air between the filament and the mesoporous ion-exchange material. The latter remains dark, since it holds water well and air does not penetrate there. The images shown in Figure 3a–f allow concluding that drying of the swollen membrane begins at the intersections of the reinforcing filaments and then spreads along the filaments through the entire membrane. The pores between the filament and ion-exchange material were also detected in images of scanning microscopy of the heterogeneous MK-40 and MA-41 membrane [69].

Similar images are obtained in the case of the CJMC-3 membrane. However, in the case of the CMX membrane (Figure 3d–f), drying occurs fairly evenly over the surface and it is not possible to visualize the “rapid water evaporation” spots, although these membranes contain reinforcing fabric as well. Apparently, this is due to the fact that both the reinforcing cloth and the inert filler of the CMX ion-exchange composite material are made of the same PVC polymer, that is, they have good adhesion [44,70]. 

In the following sections, we show that these features of the geometry and structure of the CJMC-3 and CJMC-5 membranes play a significant role in their transport and electrochemical characteristics.

### 3.2. Transport Characteristics

*Electrical conductivity.* The concentration dependences of the electrical conductivity of the studied membranes in NaCl, CaCl_2_, and Na_2_SO_4_ solutions are shown in Figure 4. The structural and transport parameters of the membranes, which were found from these concentration dependences using the microheterogeneous model [61], are summarized in Table 2. In the vicinity of the isoconductivity point characterized by the concentration $C_{iso}$, where the electrical conductivities of the external solution, *κ**, and its gel phase, $\bar{\kappa }$ are equal, the CMX membrane shows the highest conductivity among the studied membranes. This result is expected because the value of $\bar{\kappa }$ is directly proportional to the ion-exchange capacity of the gel phase, $\bar{Q}$, the diffusion coefficient of the counterion in the gel phase, $\bar{D_{i}}$, and the charge of the counterion, $z_{i}$:(7)$$\bar{\kappa }=\frac{z_{i}{\bar{D}}_{i}\bar{Q}F^{2}}{RT}$$

Equation (7) was obtained in the framework of the microheterogeneous model [61] under the assumption that the presence of coions in the gel phase can be neglected. 

Note that the difference in the conductivity values of CMX, on the one hand, and CJMC-3 and CJMC-5, on the other hand, is not as large as might be expected considering the values $\bar{Q}$ of the studied membranes (Table 2). Apparently, the more porous structure of the ion-exchange material removes some steric difficulties that arise during the transport of ions in the highly cross-linked ion-exchange material of CMX. This is evidenced by a comparison of the values of the diffusion coefficient of Na^+^ ions in the gel phase of the membranes normalized to its values in the free solution (at infinite dilution), Table 2.

According to Equation (8) [61],
(8)$$\kappa^{*}={\bar{\kappa }}^{f_{1}}\kappa^{f_{2}}$$
which holds true in the vicinity of the isoconductivity point (0.1$C_{iso}$ < C< 10$C_{iso}$), the electrical conductivity of the membranes is dependent not only on the electrical conductivity of the gel phase, but also on the electrical conductivity of the solution in the intergel spaces (identical to the external solution). The more the concentration of the external solution differs from $C_{iso}$ and the larger the volume fraction of the intergel spaces, $f_{2}$, the more significant this effect. The CJMCED membranes have significantly high $f_{2}$ values compared to CMX (Table 2), due to the porous structure of the ion-exchange material and the presence of macropores at the interfaces of the reinforcing thread and ion-exchange material. Therefore, these membranes demonstrate higher conductivity than CMX in solutions with the electrolyte concentration significantly exceeding $C_{iso}$, which is in the range 0.02–0.06 eq L^−1^.

A slight decrease in the electrical conductivity of all the studied membranes is observed when a doubly charged coion (CEM/Na_2_SO_4_ system) replaces a singly charged coion (CEM/NaCl system). This well-known phenomenon is due to the stronger Donnan (electrostatic) exclusion of the doubly charged SO_4_^2−^ coion than the singly charged Cl^−^ coion [72], that is, this phenomenon is mainly due to a decrease in the coion’s contribution to the membrane conductivity. 

A sharp decrease in the electrical conductivity of the studied CEMs while transitioning from a NaCl solution to CaCl_2_ is expected and described for many membranes with sulfonate fixed groups [72,73,74,75,76]. Among the reasons for this phenomenon, the following can be distinguished: inhibition of doubly charged counterions as a result of ion–ion interactions with two fixed groups simultaneously [72,77]; steric difficulties in transporting large, highly hydrated calcium ions [74,76]; the formation of weakly dissociating ion-ion associates “sulfo group-calcium ion”. According to [78,79], the formation of weakly dissociating associates of multiply charged counterions (such as calcium, magnesium, or aluminum) with sulfonate groups reduces their ionization, which is equivalent to a decrease in the effective ion-exchange capacity of the membrane. The phenomena listed above lead to a decrease in the diffusion coefficient of the calcium ion in the gel phase of the membrane, which can be easily estimated using an expression derived from Equation (7)
(9)$$\frac{{\bar{\kappa }}_{NaCl}}{{\bar{\kappa }}_{CaCl_{2}}}=\frac{{\bar{D}}_{Na^{+}}}{2{\bar{D}}_{Ca^{2+}}}$$
where 2 is the charge number of Ca^2+^, $z_{Ca^{2+}}$. The determined values for the ratio of the diffusion coefficients of Na^+^ and Ca^2+^ ions are summarized in Table 2. In all cases, the values ${\bar{D}}_{Na^{+}}/{\bar{D}}_{Ca^{2+}}$ exceed the ratio of diffusion coefficients of counterions in the solution $D_{Na^{+}}/D_{Ca^{2+}}$ = 1.7 (for evaluations, we used the values of diffusion coefficients at infinite dilution $D_{Na^{+}}$ = 1.334 × 10^−5^ and $D_{Ca^{2+}}$= 0.792 × 10^−5^ cm^2^ s^−1^ [55]). However, in CJMC-3 and CJMC-5, the inhibition of calcium counterions in the gel phase of the membranes is not as significant as in the case of CMX. The reason for this phenomenon, which is certainly positive for the further use of CJMCED in the ED of calcium-containing solutions, is likely to be a lower concentration of sulfonate groups (Table 1 and Table 2) and, accordingly, a smaller number of associates, which these groups can form with calcium counterions. Another reason is the larger pores of the CJMC-3 and CJMC-5 ion-exchange material, in which the Ca^2+^ ions do not experience steric hindrance. The presence of more “spacious” pores is reflected by the value of the parameter *f*_2_, which is more than three times higher for CJMC-3 and CJMC-5 than for the CMX membrane (Table 2).

*Diffusion permeability*. Figure 5 shows the concentration dependences of the integral coefficient of diffusion permeability, P, of the studied membranes in NaCl and CaCl_2_ solutions. The P values for these electrolytes in both CJMC-3 and CJMC-5 are an order of magnitude or higher than the measured integral diffusion permeability coefficients in the CMX membrane. Most likely, the dominant role in the case of CJMCED is played by electrolyte diffusion through macropores that are absent in the CMX membrane. Replacing the singly charged counterion (CEM/NaCl system) with the doubly charged counterion (CEM/CaCl_2_ system) is accompanied by an increase in the diffusion permeability of the membranes. This result is predictable. It is explained by increased sorption of the electrolyte due to two reasons: an increase in the attractive force between the cation and the doubly charged counterion [72], as well as the formation of weakly dissociating associates of Ca^2+^ ions with sulfonate groups, which reduce the effective ion-exchange capacity of the membrane [79].

In addition to electrostatic interactions, which are determined by the Donnan relations, interactions that are more complex are possible. They are determined by the hydration degree of the transported ions and fixed groups, by the membrane matrix material, and are discussed in the review [80].

It is noteworthy that in CEM/CaCl_2_ systems, the slope of the dependence P vs. C in the case of CaCl_2_ is less than in the case of NaCl (Figure 5). In the case of CJMC-3, at C > 0.4 eq L^−1^, a decrease in P is observed with an increase in C. Such a trend of the dependence P vs. C is also observed in the case of the MA-41/Na_2_SO_4_ [64]. It is explained by the fact that the diffusion coefficient of Na_2_SO_4_ in the solution decreases markedly with increasing concentration of this electrolyte [64]. Since electrolyte diffusion in heterogeneous membranes occurs mainly through the intergel spaces filled with an electrically neutral solution, the indicated dependence *D*_Na_2_SO_4__ vs. C contributes to the trend of the dependence P vs. C. However, in the case of CaCl_2_, the diffusion coefficient of the electrolyte in the solution decreases only at low concentrations, reaching a minimum at 0.2–0.4 eq L^−1^, and then increases with increasing C [81]. Therefore, the reason for the decrease in P with increasing C, which is valid for Na_2_SO_4_, cannot be applied in the case of CaCl_2_. The main reason for the observed trend is apparently a decrease in the water content in the IEM with an increase in the concentration of the external solution. This results from a decrease in the osmotic pressures difference between the solution and the membrane [73,82]. The difference in osmotic pressures is the “driving force” of membrane swelling and its decrease leads to a decrease in pore size and their water content. This effect should be especially notable in the weakly crosslinked membranes CJMC-3 and CJMC-5. A decrease in the water content in the pores leads to an increase in the concentration of fixed groups per volume of sorbed water. As follows from the Donnan relation, this should cause a more significant electrostatic exclusion of coions and, as a consequence, a decrease in membrane permeability with respect to electrolyte diffusion. Freeman et al. [77,79] investigated this effect in detail for gel IEMs. Another reason for the decrease in P with increasing C may be an increase in steric hindrance with a decrease in pore size, which can be especially manifested during the transport of large highly hydrated Ca^2+^ ions. The second reason is less likely than the first, since the diffusion of the electrolyte in the IEM is controlled by coions. The discovered effect requires additional investigations, which are planned to be conducted in the future.

As for the membrane CMX, our study and similar [76] studies show that with an increase in the concentration of the CaCl_2_ solution from 0.02 to 2 eq L^−1^, a decrease in the water content of this rather strongly cross-linked membrane is comparable to the measurement error (5%). Thus, “shrinking” of the ion-exchange matrix with increasing concentration of the external solution practically does not affect the concentration dependences of the diffusion permeability of CMX.

*Counterion transport numbers.* The (electromigration) ion transport number is the fraction of electric charge carried by a given ion under conditions when only an electric force is applied as a driving force; $t_{1}^{*}$ characterizes the counterion permselectivity of the membrane [83]. The concentration dependences of the counterions transport numbers of $t_{1}^{*}$, which are determined according to Equation (4) from the values of electrical conductivity and coefficient of diffusion permeability, are presented in Figure 6. In the case of the CMX/NaCl system, the obtained values of $t_{1}^{*}$ are close to the results [64] obtained for the CM2 membrane with a similar structure.

For all studied membranes, the transport numbers decrease when replacing NaCl solution with CaCl_2_ solution. The CMX membrane has higher values of $t_{1}^{*}$ compared to CJMC-3 and CJMC-5. Due to slightly higher conductivity (Figure 4b,c) and lower diffusion permeability (Figure 5b,c), CJMC-5 is characterized by considerably greater counterion transport numbers in comparison to the CJMC-3 membrane (Figure 6a,b).

As discussed above, the decrease in $t_{1}^{*}$ when changing NaCl for CaCl_2_, is apparently caused by electrostatic interactions of Ca^2+^ ions with the sulfonate fixed groups of the studied membranes. The lower permselectivity of the CJMC-3 and CJMC-5 membranes is due to (a) the lower ion-exchange capacity (Table 1 and Table 2) and (b) larger pores, in particular, extended macropores (Figure 3), which are absent in the CMX membrane. Large macropores and structural defects are filled with the electrically neutral solution, where ionic transport is not selective.

It should be noted that low counterion permselectivity of the CJMC-3 and CJMC-5 membranes occurs only in rather concentrated NaCl and CaCl_2_ solution. This means that these membranes should not be used in electrodialyzers-concentrators. However, in solutions whose total dissolved solids (TDS) does not exceed 5 g L^−1^ (≈ 0.1 mol L^−1^) (solution TDS corresponding to the case where ED has economic advantages over reverse osmosis [84]), the true transport numbers of counterions in CJMC-3 and CJMC-5 reach the values of 0.99 (NaCl) and 0.97 (CaCl_2_). Thus, use of CJMC-3 and CJMC-5 in the ED of relatively dilute solutions (brackish water) seems very attractive. The counterion transport numbers in these membranes are rather close to the values obtained for some heterogeneous (MK-40, JSC Shchekinoazot, Russia) [73] and homogeneous (CEM Type-I, Fijifilm, The Netherlands) [44] ion-exchange membranes, which are actively used for the ED of relatively dilute solutions.

Table 3 compares the “true” counterion transport numbers, $t_{1}^{*}$, found from the conductivity and diffusion permeability measurements, Equation (4), and the “apparent” transport numbers, ($t_{1app}^{*}$), determined using the potentiometric method [38,85]. In the last case, the potential difference between two reversible to the anion electrodes separated by the membrane under study, E, is measured. The concentration of the electrolyte solution bathing the membrane from one side is $C_{1}$, and that from the other side, $C_{2}$. In the case of 1:1 electrolyte,
(10)$$t_{1app}^{*}=\frac{E}{E^{0}}$$
where $E^{0}=\frac{RT}{F}ln\frac{C_{1}}{C_{2}}$ is the maximum possible value of E achieved for a perfectly selective membrane. As a rule, the catalogs of manufacturers and the most scientific papers give the apparent transport numbers, which are easier to find. The relationship between the “true” ($t_{1}^{*}$) and the apparent ($t_{1app}^{*}$) transport numbers is given by the Scatchard equation [56,85]
(11)$$t_{1app}^{*}=t_{1}^{*}-z_{1}n_{1}m_{s}M_{w}t_{w}$$
where $z_{1}$ and $n_{1}$ are the charge and stoichiometric numbers of counterion, $m_{s}$ is the molality of the solution, $M_{w}$ is the molar mass of water, and $t_{w}$ is the water transport number. The last value shows how many moles of water are transported when one Faraday of electric charge passes through the membrane. The value of $t_{1app}^{*}$ depends on the “true” transport number $t_{1}^{*}$, and on the value of $t_{w}$ that is determined by the electroosmotic permeability of the membrane, Equation (11). Therefore, the $t_{1app}^{*}$ values are generally lower than the $t_{1}^{*}$, values, which characterize the real selectivity of the membranes in ED. This difference can be illustrated by the values presented in Table 3. The CMX membrane is denser than the CJMCED ones, the water content in mol H_2_O/mol functional groups (= 8) is the lowest; CJMC-5 has 23, and CJMC-3, 27 mol H_2_O/mol functional groups (Table 1). This order correlates well with the order of the values of $t_{w}$: CMX < CJMC-5 < CJMC-3. The permselectivity of the membrane varies in the opposite order: the higher the water content, the less the value of order $t_{1}^{*}$. The values of $t_{1app}^{*}$ change in the same order, and the difference between the values is higher than in the case of $t_{1}^{*}$. The highest difference is observed between the water transport numbers, −$t_{w}$. They change from 4 (CMX) to 16 (CJMC-3): the higher water content in mol H_2_O/mol functional groups, the higher $t_{w}$. This correlation was earlier established by Berezina et al. [56] for a series of perfluorosulfonic acid cation-exchange MF-4SK membranes (MF-4SK is a Russian analog of Nafion).

In this series, the water content varied from 9.8 to 36.5 mol H_2_O/mol functional groups caused by different membrane pretreatment. For the lowest value of water content, $t_{w}$ was about 6, and for the highest, 16 mol H_2_O/F (at the NaCl concentration close to 0.1 mol dm^−3^). Therefore, we see that our results are in a good accordance with Ref. [56]. Note also the value of $t_{w}$= 4 mol H_2_O/F found for the CMX membrane is equal to the $t_{w}$ value for a Neosepta CM2 membrane (which was the predecessor of the Neosepta CMX membrane) reported by Larchet et al. [86]. The abovementioned correlations can be explained by the relationship between the water content and the size of membrane pore (under condition that the exchange capacity is sufficiently close): the higher the water content, the greater the pore size, and the easier the transport of water and coions.

### 3.3. Current Voltage Curves and Water Splitting Rate

Figure 7 shows the CVCs of the studied membranes and the difference between the pH, ΔpH, of the outlet and inlet NaCl solution passing through the desalination compartment of the lab-scale ED cell shown in Figure 1. CVCs are normalized to the values of limiting current, $i_{lim}^{theor}$, which are calculated according to Equation (6). For calculating $i_{lim}^{theor}$ (Equation (6)), we used the value of the NaCl diffusion coefficient in an infinitely dilute solution since the density of the diffusion flux from the solution bulk to the depleted membrane surface is determined by the concentration gradient at a point near the membrane surface, where the electrolyte concentration is close to 0. The effective transport number values of counterions in the membrane, $T_{1}$, were taken as equal to 1, since at the solution concentration of 0.02 eq L^−1^, used in the experiment, the “true” transport numbers of Na^+^ ions differ from 1 by no more than 0.4%: $t_{Na}^{*}$, found for C = 0.02 eq L^−1^ by extrapolating the experimental dependence $t_{1}^{*}$ vs. *C* (Figure 6a), equals 0.999 for CMX and 0.996 for CJMC- 3 and CJMC-5. The reduced potential drop is determined using Equation (5).

All the studied membranes demonstrate close experimental values of limiting current, $i_{lim}^{exp}$. These values are similar to $i_{lim}^{theor}$, calculated in the framework of the convective-diffusion model according to Equation (6). In the case of the CJMC-5 membrane $i_{lim}^{exp}$ exceeds $i_{lim}^{theor}$ by 15%. Most likely, this excess is due to a more pronounced electrical inhomogeneity of CJMC-5 in comparison with CJMC-3 and CMX (Figure 2). The reason for this inhomogeneity is the closer proximity of the inert cloth to the membrane–solution interface in the CJMC-5 membrane than in CJMC-3. An increase in the electrical heterogeneity causes an increase in the tangential component of the electric field [88], which contributes to the emergence of equilibrium electroconvection that develops at relatively small values of potential drop at the end of initial linear section I of the CVC (Figure 7a) [89,90,91,92].

Note that the surface of CJMCED membranes is more hydrophilic than that of the CMX: the contact angles for the CJMC-3, CJMC-5 and CMX are 57 ± 2, 54 ± 2, and 49 ± 2 (degrees), respectively.

This should be due to a lower exchange capacity of the CJMCED membranes. On the other hand, the CJMCED membranes have a higher water content (Table 1), which should be determined by their more porous structure compared to the CMX membrane. Perhaps, the smaller fraction of the charge-bearing regions on the surface and a significant fraction of the surface constituted of the pore mouths open into the external solution, cause a longer inclined plateau (section II of the CVC) of the CJMCED membranes. This surface feature could be the reason for the increase in the voltage, at which the transition to the nonequilibrium electroconvection [92,93] (section III of the CVC) occurs. It should be noted that in the overlimiting current modes (section III of the CVC), the CVCs of the CJMC-3 and CJMC-5 membranes have more intensive oscillations of the potential drop than the CVCs of the CMX membrane, although this occurs at essentially higher potential drops than in the case of CMX. The observed oscillations are the result of the formation of unstable electroconvective vortices near the membrane surface [90,92]. A significant role in stimulating electroconvection can be played by inert threads of the reinforcing cloth, whose intersections are uniformly distributed near the membrane surface (Figure 2 and Figure 3) with the spacing that is of the same order of magnitude as the thickness of the diffusion layer in the DC (about 220 μm in the case of the NaCl solution) [47]. These intersections of the inert threads can increase the tangential component of the electric current, which stimulates the development of electroconvection [94,95,96,97]. To confirm these assumptions, it is necessary to conduct additional experiments. At the same time, it can already be expected that the CJMC-3 and CJMC-5 membranes can serve as an excellent substrate for the formation of surfaces with a high ability to develop electroconvection.

It should be noted that the difference in pH of the output and input solutions is determined by the difference in the intensity of water splitting on the surfaces of the CEM and the auxiliary anion-exchange membrane forming the DC. In all cases, the MA-41 membrane was used as the latter (Figure 1). When the water splitting rate is higher at the surface of the cation-exchange membrane, the solution becomes more alkaline. Otherwise, it acidifies.

As Figure 7b shows, the trend of the dependence of ΔpH vs. $i/i_{lim}$ is rather similar in all considered cases. The pH of the output solution does not change up to $i/i_{lim}$ ≈ 1.5, which corresponds to reaching the limiting current density at AEM ($i_{lim}^{Cl^{-}}=1.5i_{lim}^{Na^{+}}$, since the mobility of Cl^−^ ions is 1.5 times higher than the mobility of Na^+^ ions). Then, water splitting at the AEM begins, which supplies H^+^ ions to the DC. The water splitting rate at the CEM is significantly lower, which determines the decrease in the pH of the output solution. Nevertheless, it can be noted that in the case of the CJMC-5 membrane, the water splitting rate at sufficiently high currents is higher than in the case of the CJMC-3 membrane. This may be the reason for weaker electroconvection of CJMC-5 compared to CJMC-3. It is known [98,99,100] that the high rate of water splitting partially suppresses electroconvection, since ions H^+^ (or OH^−^) entering an extended space charge region near the membrane reduce the value of the space charge, which in turn reduces electroconvection [89,101,102].

### 3.4. Fouling with Aromatic Macromolecular Substances

The presence of macropores in CJMC-3 and CJMC-5 in combination with the aliphatic polymer matrix, which contains small quantities of aromatic cross-linking substances, makes these membranes promising for use in ED processing of solutions that contain high-molecular-weight aromatic substances. It is known [31,103] that it is fouling of ion-exchange membranes by such substances as anthocyanins, proanthocyanins, and tannins that restrains the widespread use of electrodialysis in the conditioning of wine [17], pH adjustment of juice and wine [104], and the extraction of antioxidants from food industry waste [105], etc. Therefore, researchers make great efforts to tackle this phenomenon [106,107,108] and use complex mixtures of organic solvents or inorganic oxidizing agents to clean membranes [107,109].

Prior to the experiments, the samples were kept in a buffer solution of pH 3.56, for 2 h to impart a sufficiently bright and well-detectable color to natural dyes (mainly anthocyanins), which are sorbed by ion-exchange membranes [103].

Our experiments show that natural dyes (anthocyanins and other substances), which are contained in cranberry juice, easily penetrate the CJMC-3 and CJMC-5 membranes after several hours of contact with the juice (Figure 8a,b). This is evidenced by the complete staining of the cross-sections of these membranes, while in the case of CMX, these substances are mainly localized on the surface and in near- surface layers (Figure 8c).

However, after a longer time (more than 100 h) of soaking CMX, CJMC-3, and CJMC-5 in cranberry juice, differences in coloring of the surface and cross-sections of these membranes caused by high-molecular-weight aromatic substances almost disappear (Figure 9a). It is important to note that natural aromatic dyes are almost completely removed from the CJMC-3 and CJMC-5 membranes by soaking them in the 1 M NaCl solution for 24 h (Figure 9b). At the same time, the color of the CMX membrane (hence the quantity of dyes in it) does not undergo major changes. The behavior of the CJMC-5 membrane in contact with high molecular weight aromatic substances is very similar to that of the CJMC-3 membrane; the photographs illustrating the process of sorption and extraction of the dye in the case of the CJMC-5 membrane practically do not differ from those for the CJMC-5 membrane and for this reason are not shown here.

Apparently, the main reason for the difference in the behavior of CJMCED and CMX membranes, with respect to interaction with solutions containing high-molecular-weight aromatic substances, is the chemical nature of their matrix. For CJMCED membranes, which have a predominantly aliphatic matrix, the removal occurs to a greater extent than for CMX membranes because of the aromatic matrix of the latter. In this case, relatively strong π–π stacking interactions are possible, which leads to a stronger attraction of natural dye molecules to the matrix of the CMX membrane. In addition, the larger pores of the CJMCED membranes facilitate the extraction process.

These preliminary experiments suggest that the use of CJMC-3 and CJMC-5 in ED processing of liquid media of the food industry will contribute to counteracting membrane fouling caused by high-molecular-weight aromatic dyes. In addition, they may find application in ED processing of dilute solutions of other aromatic substances, for example, aromatic amino acids, fulvic acids, pesticides, etc.

## 4. Conclusions

The characterization of the CJMC-3 and CJMC-5 cation-exchange membranes have shown that the conductivity, CVC curves (at relatively low voltages) and water splitting rate (characterized by ΔpH) for both CJMCED are rather close to these characteristics for the CMX membrane. Nevertheless, the electrolyte diffusion permeability of the CJMCED membranes is significantly higher than that of the CMX membrane. It is explained by the fact that the pores of the CJMCED membranes are significantly more spacious than those in the CMX membrane. It seems that the size of mesopores in the CJMCED membranes is higher than in the CMX; in addition, the CJMCED membranes have macropores, while the CMX membrane does not. This structural feature of the CJMCED membranes results in certain advantages and disadvantages. The lower resistance of these membranes to the transport of multiply charged and large ions allows us to recommend them for use in ED treatment of natural and industrial waste waters for the removal of large ions. These membranes can also be applied in the treatment of food solutions, such as juices, wine, milk products, etc. At the same time, the large pores result in lower counterion permselectivity, such that the CJMCED membranes can hardly be used in ED concentration of solutions.

## Figures and Tables

**Figure 1 membranes-10-00165-f001:**
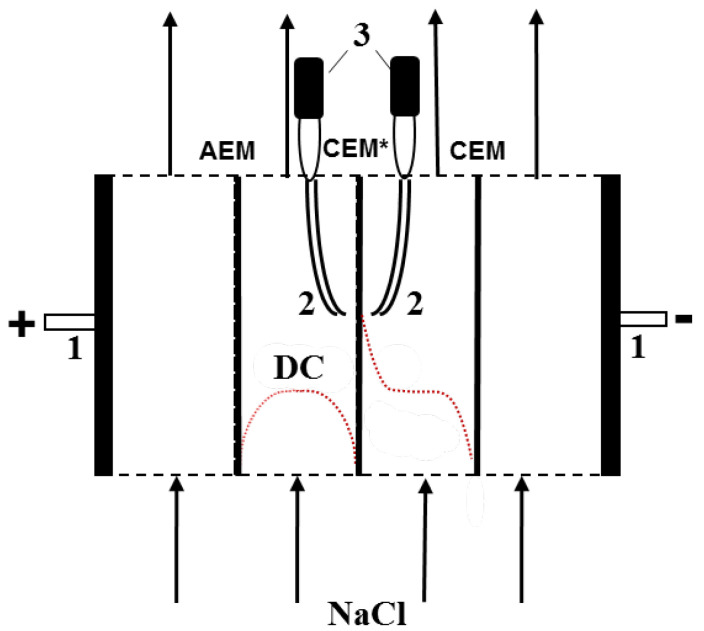
Schematic diagram of the electrochemical cell for measuring the current–voltage characteristics and pH at the inlet and outlet of the DC formed by the cation-exchange membrane under study (CEM*) and an auxiliary anion-exchange membrane: 1—polarizing platinum electrodes; 2—Luggin capillaries connected to the microreservoir in which Ag/AgCl electrodes are immersed (3). The dashed lines show the concentration profiles of the electrolyte in the cell channels separated by the membrane under study.

**Figure 2 membranes-10-00165-f002:**
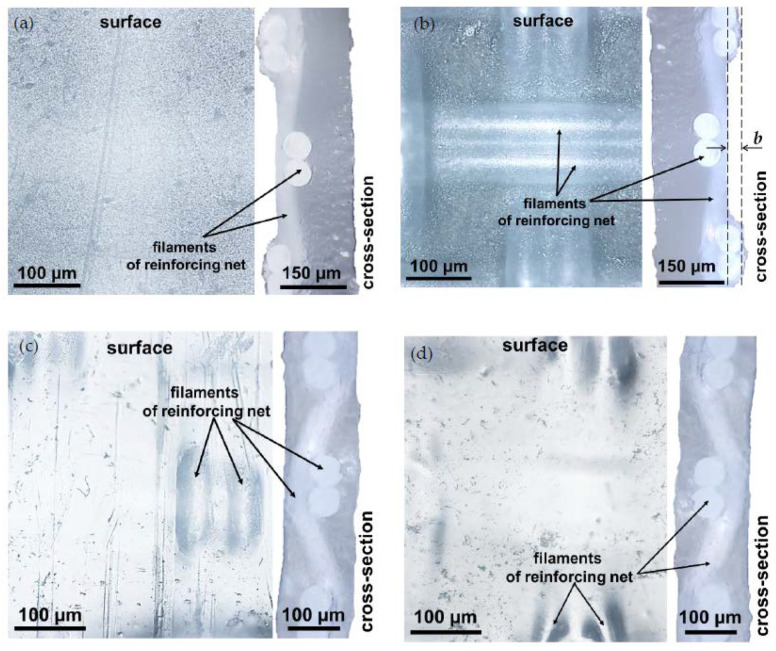
Optical images of the surface and cross sections of CJMC-3 (**a**,**b**) and CJMC-5 (**c**,**d**) membranes. Surface I is shown in figures (**a**,**c**); surface II is shown in figures (**b**,**d**). The explanations are in the text.

**Figure 3 membranes-10-00165-f003:**
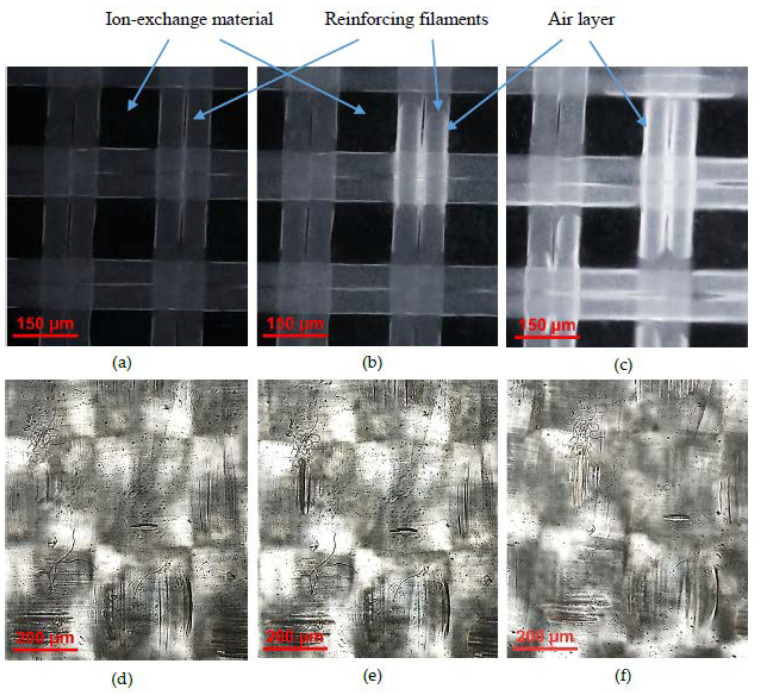
Optical microscope images with transmitted light of the surface of initially swollen CJMC-5 (**a**–**c**) and CMX (**d**–**f**) membranes after their contact with air for 5 s (**a**,**d**), 20 s (**b**,**e**), and 150 s (**c**,**f**).

**Figure 4 membranes-10-00165-f004:**
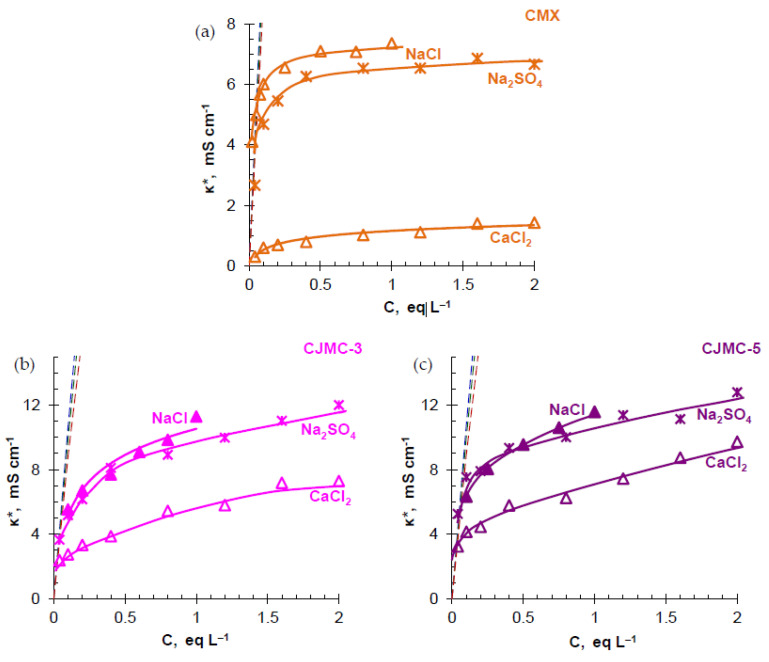
Electrical conductivity of CMX (**a**), CJMC-3 (**b**), and CJMC-5 (**c**) membranes vs. concentration of NaCl, CaCl_2_, and Na_2_SO_4_ in the external solution. The dashed line shows the concentration dependence of the NaCl (blue), Na_2_SO_4_ (red), CaCl_2_ (green) solution conductivity. The continuous lines are for the guide of eyes.

**Figure 5 membranes-10-00165-f005:**
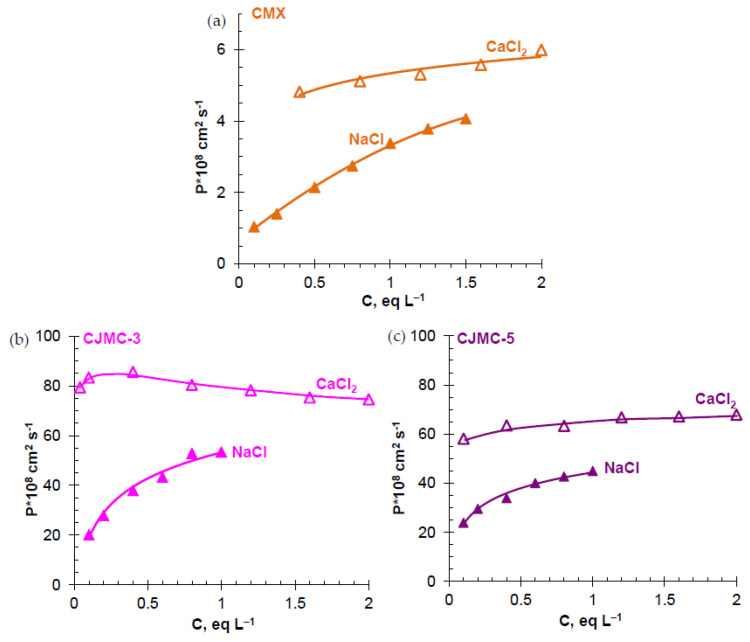
Concentration dependences of the integral diffusion permeability coefficient of CMX (**a**), CJMC-3 (**b**), and CJMC-5 (**c**) membranes vs. concentration of NaCl and CaCl_2_ solutions. The lines are for the guide of eyes.

**Figure 6 membranes-10-00165-f006:**
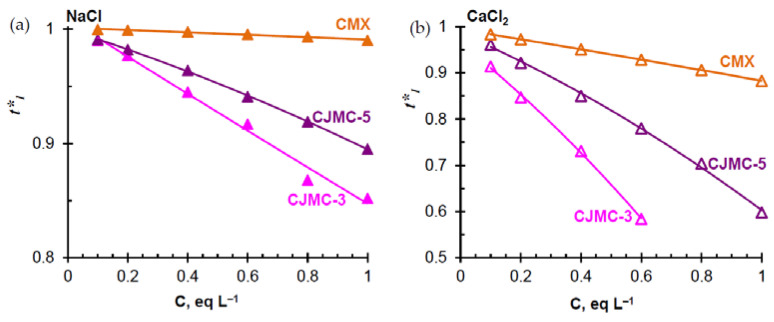
The counterion transport numbers in the studied cation-exchange membranes vs. concentration of NaCl (**a**) and CaCl_2_ (**b**) solutions. The lines are drawn to guide the eye.

**Figure 7 membranes-10-00165-f007:**
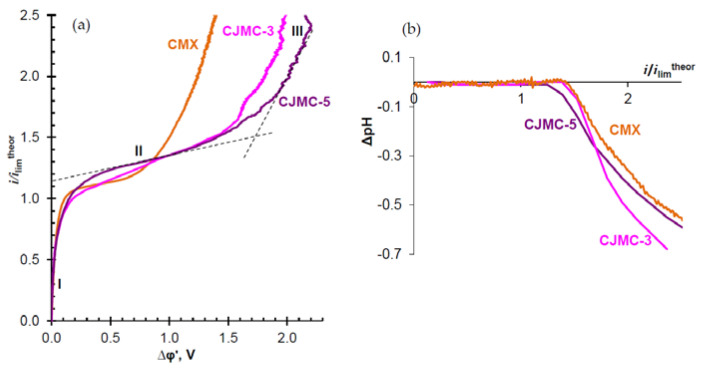
Current–voltage curves (CVCs) of the studied membranes (surface I) (**a**) and differences between the outlet and inlet pH of the NaCl solution passing through the desalination compartment (**b**). The desalination channel is formed by one of the CEMs under study and a MA-41 anion-exchange membrane. The concentration of the solution at the inlet of the DC and other channels of the electrodialysis cell is 0.02 eq L^−1^.

**Figure 8 membranes-10-00165-f008:**
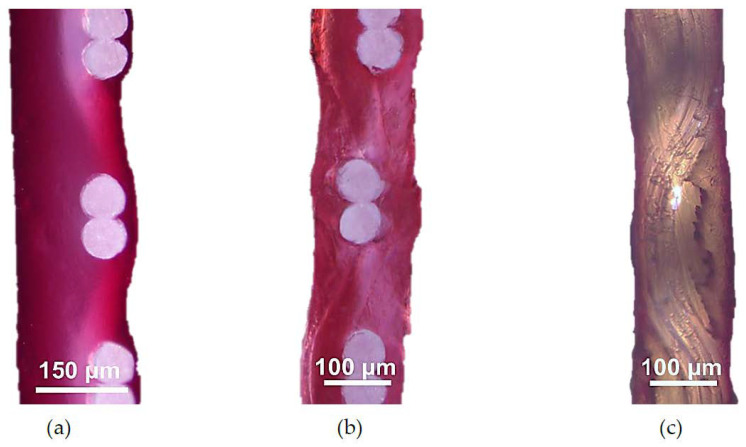
Optical images of the cross sections of CJMC-3 (**a**), CJMC-5 (**b**), and CMX (**c**) samples after soaking in cranberry juice for 2 h.

**Figure 9 membranes-10-00165-f009:**
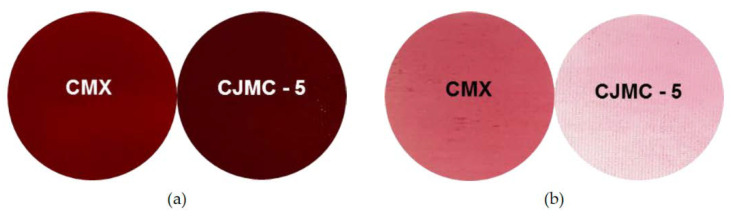
Photos of CMX and CJMC-5 samples after soaking in cranberry juice for 168 h (**a**) and subsequent soaking in 1 M NaCl solution for 24 h (**b**) for desorption of dyes.

**Table 1 membranes-10-00165-t001:** Characteristics of swollen (sw) and dry membranes under study.

Membranes	Thickness (sw) ^1^, μm	Exchange Capacity (sw), mmol g^−1^	Water Content, g_H_2_O_/g_dry_, %	Water Content, mol H_2_O/mol Functional Groups	Resistance ^2^ (Ohm cm^2^)
CJMC-3	185 ± 5 190 ± 20 [37] 170 ± 0.01 [53]	0.63 ± 0.05 0.80–1.0 [37] 0.80–1.0 [53]	44 ± 3 40–45 [37] 35–45 [53]	27 ± 1	2.2 ± 0.3 3.0 ± 0.5 [37] 2.5–3.5 [53]
CJMC-5	140 ± 3 140 ± 3 ^3^	0.57 ± 0.07 1.00–1.20 ^3^	32 ± 5 25–27 ^3^	23 ± 1	1.4 ± 0.2 2.0–2.5 ^3^
CMX [44]	170 ± 5 164 [54]	1.61 ± 0.05 1.62 [54]	28 ± 3 18 [54]	8 ± 1 9 [28]	2.6 ± 0.3 2.91 [54]

^1^ Membrane equilibrated with 0.02 M NaCl solution; ^2^ Membrane equilibrated with 0.5 M NaCl solution; ^3^ The data were provided by the manufacturer.

**Table 2 membranes-10-00165-t002:** The values of the electrical conductivity at the isoconductivity point, $\bar{\kappa }$, volume fraction, $f_{1}$, and exchange capacity of the gel phase, $\bar{Q}$, of the investigated membranes.

Membranes	$\bar{\kappa }$, mS cm^−1^			$\bar{Q}$, mmol cm^−3^_sw gel_	$f_{1}$			$\frac{{\bar{D}}_{Na^{+}}}{{\bar{D}}_{SO_{4}^{2-}}}$	$\frac{{\bar{D}}_{Na^{+}}}{{\bar{D}}_{Ca^{2+}}}$
	NaCl	CaCl_2_	Na_2_SO_4_		NaCl	CaCl_2_	Na_2_SO_4_		
CMX	5.5 ± 0.2 6.7 [71]	0.3 ± 0.3	4.9 ± 0.3	1.8 ± 0.2	0.11	0.22	0.11	2	37
CJMC-3	3.9 ± 0.3	1.3 ± 0.3	3.7 ± 0.3	0.9 ± 0.2	0.33	0.38	0.35	2	6
CJMC-5	5.2 ± 0.3	2.4 ± 0.3	4.8 ± 0.3	0.8 ± 0.2	0.28	0.33	0.32	2	2

**Table 3 membranes-10-00165-t003:** The values of the “true” ($t_{1}^{*}$) and apparent ($t_{1app}^{*}$) counterion transport numbers and the water transport number ($t_{w}$) in the studied membranes.

Membranes	$t_{1}^{*}$	$t_{1app}^{*}$	*t_w_* mol H_2_O/F
CJMC-3	0.982 (0.15 M NaCl)	0.94 ^1^ [37]	16
CJMC-5	0.989 (0.15 M NaCl)	0.96 ^1^	11
CMX	0.997 (0.25 M NaCl)	0.98 ^2^ [87]	4

^1^ Found from the membrane potential measured between 0.1 mol/L KCl and 0.2 mol/L KCl solution. ^2^ Found from the membrane potential measured between 0.1 mol/L NaCl and 0.5 mol/L NaCl solution.

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
