# Peer review of "Transport and Electrochemical Characteristics of CJMCED Homogeneous Cation Exchange Membranes in Sodium Chloride, Calcium Chloride, and Sodium Sulfate Solutions"

_membranes, 2020, doi:10.3390/membranes10080165_

Round 1

Reviewer 1 Report

This manuscript presents a fundamental study on the transport and electrochemical properties of homogeneous cation-exchange membranes developed by Hefei Chemjoy Polymer Material Co. Ltd. Bench-scale experiments were performed with different salt solutions. Convincing explanations were provided to support experimental observations. The content presented in this work is novel and without doubt of interest to readers. This study could potentially serve as a guide in determining possible areas of applications for the studied membranes. Hence, I recommend the publication of this manuscript after the authors have addressed the following comments/suggestions:

This manuscript generally reads fine but I strongly recommend that the authors further improve on its grammar and overall style.

Line 18, …CJMC–3 and CJMC–5 cation-exchange membranes (CJMCED) are characterized. The authors only used CJMCED once in the main body but numerous times in the title and abstract. Please use CJMCED more consistently in the main body or consider removing CJMCED from the title.

Line 19, ‘in respect to’ is not standard English. It should be either ‘with respect to’ or ‘in respect of’.

Line 20, current-voltage (I-V) curves.

Line 29, the word ‘perspectives’ is incorrectly used. Please substitute it with another word.

Line 37, and bioelectrochemical systems. Citation(s) please.

Line 54, selective ED and ED metathesis

Introduction section, please provide a short description of the working principles of the electrodialysis process. Please also include a writeup on CEMs (e.g., what are CEMs, differences between homogeneous and heterogeneous) and why knowledge of the structure, transport, and electrochemical properties are important.

Line 62, development of IEMs

Line 73, current-voltage curves (CVC). The authors used I-V curves in the abstract and in the conclusions but CVC in the main body. Please be consistent.

Please use abbreviations (e.g., CEM, IEMs, ED, and etc) consistently throughout the manuscript.

Table 1, very confusing. It is hard to figure out the numbers that represent the respective membranes.

Lines 103 and 104, diffusion coefficients: 1.61*10-5, 1.34*10-5, 1.23*10-5. Please check the rest of the manuscript for similar errors.

Line 413, typo. (b) instead of (c).

Figure 7(a), the label ‘CJMC-5’ is incomplete.

Lines 427 and 428, the authors mentioned that the CJMC membranes are more hydrophilic but the contact angles reported suggest otherwise.

Figure 9, where are photos of CJMC-3 membranes?

Section 3.4, please explain why dyes can be more easily removed from CJMCED membranes as compared to CMX membranes. Do the transport and electrochemical characteristics of CJMCED membranes remain the same after membrane cleaning? Please provide evidence.

Including a list of nomenclature will be useful to potential readers.

Author Response

Dear Reviewer,

Thank you for your time and efforts. We greatly appreciate your very helpful comments and suggestions.

  1. Line 18, …CJMC–3 and CJMC–5 cation-exchange membranes (CJMCED)are characterized. The authors only used CJMCED once in the main body but numerous times in the title and abstract. Please use CJMCED more consistently in the main body or consider removing CJMCED from the title.

Authors’ response:

Thank you for this remark. Indeed, «CJMCED» is general designation of commercial IEMs (produced by Hefei Chemjoy Polymer Materials Co. Ltd.) for desalination application (Wu, L., Wang, H., Xu, T.-W. & Xu, Z.-L. Polymeric membranes, in Membrane-Based Separations in Metallurgy 297–334 (Elsevier, 2017). doi:10.1016/B978-0-12-803410-1.00012-8.) Relevant changes have been made.

  1. Line 19, ‘in respect to’ is not standard English. It should be either ‘with respect to’ or ‘in respect of’.

Authors’ response:

Thanks for this remark. We made the needed corrections (page 1).

  1. Line 20, current-voltage (I-V)

Authors’ response:

Thanks again. We replaced it in the revised manuscript with CVC.

  1. Line 29, the word ‘perspectives’ is incorrectly used. Please substitute it with another word.

Authors’ response:

We have chosen a more appropriate word: “The new membranes are promising for the use in electrodialysis demineralization of brackish water and natural food solutions.”

  1. Line 37, and bioelectrochemical systems. Citation(s) please.

Authors’ response:

Thanks for this suggestion. In the revised manuscript we made necessary changes.

  1. Line 54, selective EDand ED metathesis

Authors’ response:

Thank you, the abbreviations you proposed is now used in the manuscript.

  1. Introduction section, please provide a short description of the working principles of the electrodialysis process. Please also include a writeup on CEMs (e.g., what are CEMs, differences between homogeneous and heterogeneous) and why knowledge of the structure, transport, and electrochemical properties are important.

Authors’ response:

Thank you for this suggestion. We followed it and added two fragments in page 1 and page 2

“In this process, under the action of an electric field applied by two electrodes, cations are removed from the feed solution through cation-exchange membranes (CEMs) permeable nearly exclusively to cations, and the anions are removed through anion-exchange membranes (AEMs) permeable nearly exclusively to anions.”

“In this work, we report the results of a study of the transport characteristics (electric conductivity, diffusion permeability, transport numbers) and current-voltage curves of cation-exchange membranes CJMC - 3 and CJMC - 5 manufactured by Hefei Chemjoy Polymer Material Co. Ltd, which could help to determine the possible areas of their application.

Generally, CEMs (as well as the AEMs) are divided into homogeneous and heterogeneous types, according to their structure and the way of fabrication. Homogeneous membranes are conventionally produced by the “paste method”. They contain a homogeneous at the nanoscale level (up to 100 nm) single-phase ion-exchange matrix, which can be a single solid electrolyte (such NafionTM, DuPont Co., Wilmington, CA, USA; MF-4SK, JSC NPO Plastpolymer, Saint Petersburg, Russia) or include reinforcing fabric (some kinds of NafionTM membranes (Nafion™ N-324, Nafion™ N424 and others), Neosepta AMX, CMX, Astom, Tokyo, Japan; CJMCED and CJMAED, Hefei ChemJoy Polymer Materials, Hefei, China; and others). Heterogeneous IEMs consist of micrometer-sized (from 5 to 50 microns) ion-exchange polymer particles incorporated into an inert binder (MK-40, MA-40, Shchekinoazot, Russia; Ralex MH-PES, MEGA a.s., Czech Republic; FTAM-E, FTAM-A, FuMA-Tech GmbH, Ludwigsburg, Germany) [44]. In homogeneous IEMs, the functional charged groups are chemically bonded to the matrix, which makes a single phase extended throughout the entire membrane; in heterogeneous membranes, the charged groups are chemically bonded to the matrix, which size is however limited by the size of a resin particle, different particles are physically mixed with the inert binder [45]. Therefore, in homogeneous membranes, one polymer fulfils two function: the ion-exchange capability and the structural support; in heterogeneous membranes, these two functions are divided between two different polymers [46]. Competition between homogeneous and heterogeneous IEMs lasts for decades. Heterogeneous membranes are always thicker and consume more energy. Their greater pores conditions lower counterion permselectivity. Low conductive surface area fraction of these membranes causes higher concentration polarization, hence, greater voltage and water splitting rate at a same average current density [47]. On the other hand, heterogeneous IEMs are less costly and sometimes more chemically stable.

The aim of this study is the assessment of some properties of the recent CJMCED membranes and comparison of them with those of a well-established commercial Neosepta СМХ membrane. Although the CJMCED membranes are related to the same class as the CMX membrane (homogeneous with reinforcing fabric), the cost of CJMCED membranes is rather close to that of heterogeneous membranes. Besides, these membranes are more porous, which has its advantages and disadvantages. ”

  1. Line 62, development of IEMs

Authors’ response:

Thank you, the abbreviation you proposed was used in the manuscript.

  1. Line 73, current-voltage curves (CVC). The authors used I-V curves in the abstract and in the conclusions but CVC in the main body. Please be consistent.

Authors’ response:

Thank you for this remark. Relevant changes have been made.

  1. Please use abbreviations (e.g., CEM, IEMs, ED, and etc) consistently throughout the manuscript.

Authors’ response:

Thank you for this remark. We made necessary corrections.

  1. Table 1, very confusing. It is hard to figure out the numbers that represent the respective membranes.

Authors’ response:

Thanks for this suggestion. We have corrected Table 1. Hope, now it is clear.

  1. Lines 103 and 104, diffusion coefficients: 1.61*10-5, 1.34*10-5, 1.23*10-5. Please check the rest of the manuscript for similar errors.

Authors’ response:

Thank you for this remark. We use now the presentation of the following kind: 1.61×10-5.

  1. Line 413, typo. (b)instead of (c).

Authors’ response:

Thank you for the typo found. We replaced “(c)” with “(b)”.

  1. Figure 7(a), the label ‘CJMC-5’is incomplete.

Authors’ response:

Thanks for this remark. It is corrected.

  1. Lines 427 and 428, the authors mentioned that the CJMC membranes are more hydrophilic but the contact angles reported suggest otherwise.

Authors’ response: Thank you very much for this remark. We have corrected the error. Here is the revised fragment of the text in page 15:

“Note that the surface of CJMCED membranes is slightly more hydrophobic than that of the CMX: the contact angles for the CJMС – 3, CJMС – 5 and CMX are 57±2, 54±2 and 49±2 (degrees), respectively. This should be due to a lower exchange capacity of the CJMCED membranes. On the other hand, the CJMCED membranes have a higher water content (Table 1), which should be determined by their more porous structure compared to the CMX membrane. Perhaps, the less fraction of the charge-bearing regions on the surface and a significant fraction of the surface constituted of the pore mouths open into the external solution, cause a longer inclined plateau (section II of the CVC) of the CJMCED membranes. This surface feature could be the reason for the increase in the voltage, at which the transition to the nonequilibrium electroconvection [92,93] (section III of the CVC) occurs.”

  1. Figure 9, where are photos of CJMC-3 membranes?

Authors’ response: we added the following phrase in page 16:

“The behavior of the CJMC - 5 membrane in contact with high molecular weight aromatic substances is very similar to that of the CJMC - 3 membrane; the photographs illustrating the process of sorption and extraction of the dye in the case of the CJMC - 5 membrane practically do not differ from those for the CJMC - 5 membrane and for this reason are not shown here.”

  1. Section 3.4, please explain why dyes can be more easily removed from CJMCED membranes as compared to CMX membranes. Do the transport and electrochemical characteristics of CJMCED membranes remain the same after membrane cleaning? Please provide evidence.

Authors’ response:

We added the following explanation, page 17:

“ Apparently, the main reason for the difference in the behavior of CJMCED and СMX membranes with respect to interaction with solutions containing high-molecular-weight aromatic substances is the chemical nature of their matrix. For CJMCED membranes, which have a predominantly aliphatic matrix, the removal occurs to a greater extent than for CMX membranes because of the aromatic matrix of the latter. In this case, relatively strong π-π stacking interactions are possible, which leads to a stronger attraction of natural dyes molecules to the matrix of the СMX membrane. In addition, the larger pores of the CJMCED membranes also facilitate the extraction process.”

  1. Including a list of nomenclature will be useful to potential readers.

Authors’ response:

A list of nomenclature has been added to the manuscript.

Reviewer 2 Report

The manuscript deals with the characterization of two newly commercialized cation-exchange membranes in comparison with a well-known cation -exchange membrane from another company. this aim is interesting in the framework that the transport and performance of new ion exchange membranes will open applications for new membrane materials but there are a few imprecisions that should be clarified by the authors. For instance:

Why is the Neosepta CMX selected as references if the characteristics listed in Table 1 of the manuscript differ so much in general to the membranes being characterized and the functional groups apparently are different too? These should be clearly compared so that the variable relating the transport of water vs functional groups can provide real feedback.

Regarding the supplementary information, I have several curiosities, such as, how has the equation used to measure the cation exchange membrane capacity (S1) being derived?

Same commentaries can be observed as to the permeability equations in page 4 of the main manuscript, which differ to published literature in ion exchange membranes or membrane technology in general. Please revise.

What is the functionality of the setup in Figure 1 of the main manuscript and the MK-40 CEM in comparison with the CEM being tested?

line 192, what do the authors mean by "tops and bottoms of relief"? Besides the misunderstanding of the expression, there is no distance of 40 microns in Figure2  being discussed. Please revise.

Please explain the images in Figure 3. Holes of about 150 microns are more than macropores (lines 199-200). These holes look more like a mechanical support of the commercial membrane than macropores or "structural defects" as commented later upon discussing transport performance. Please revise.

What do the continuous lines in Figure 5 and 6 represent?

How are the electrolytes for the diffusion experiments selected?

What purpose do the authors mean in line 357?

Where was the pH measured? Bulk of near the surface? Please clarify.

English revision by a professional native translator is recommended.

State-of-the-art bibliography should be updated.

Author Response

Dear Reviewer,

we are grateful for your comments and suggestions, which allows an essential improvement of our paper.

  1. Why is the Neosepta CMX selected as references if the characteristics listed in Table 1 of the manuscript differ so much in general to the membranes being characterized and the functional groups apparently are different too? These should be clearly compared so that the variable relating the transport of water vs functional groups can provide real feedback.

Authors’ response:

Thank you for this question. Indeed, the explanations concerning the reasons of the choice of the reference membrane was not sufficiently clear formulated. In the revised manuscript the paragraph describing the choice of the CMX membrane as the reference one is as follows (page 2):

“For comparison, we present also the characteristics of a commercial Neosepta СМХ cation-exchange membrane produced by ASTOM Corporation (Tokyo, Japan), which is widely used in a great number of applications [28,33,40,49]. It is one of the best high-performance CEMs on the market. Like the novel CJMCED membranes, the CMX membrane is classified as homogeneous; it has functional sulfonic groups and is fabric reinforced. This membrane is made using the “paste method” [51]. The ion-exchange composite material of this membrane consists of two interpenetrating phases [52]: a sulfonated styrene–divinylbenzene cross-linked copolymer (ion-exchange material) and polyvinyl chloride, PVC, whose particle diameter does not exceed 60 nm. The reinforcing PVC fabric is introduced into the membrane at the stage of polymerization of the ion-exchange matrix.”

  1. Regarding the supplementary information, I have several curiosities, such as, how has the equation used to measure the cation exchange membrane capacity (S1) being derived?

Authors’ response:

The fragment describing the determination of the exchange capacity of CEM has been rewritten to avoid excessive detail:

The total exchange capacity (Qsw) of studied cation exchange membranes under study is determined by the static method [72]. A sample of the swollen membrane (about 1.0 g (msw)), were transformed into the H+ form by soaking in a 1M HCl. Then, after careful rinsing in deionized water, the sample was immersed in a 20.00 cm3 of a 0.1M NaCl solution to replace H+ ions by Na+; it was the kept until equilibrium (for 24 hours) in the above-mentioned solutions periodically shaking. After that, the concentration of the released H+ ions was determined using the potentiometric titration with 0.1 M NaOH. Titration was performed using EasyPlusTitrators (METTLER TOLEDO), with the output of the titration results data to a computer.

The calculation of the membrane total exchange capacity per weight of swollen membrane, Qsw (mmol gsw-1), was carried out by equation:

(S1)

where VT is the volume of 0.10 М NaOH solution, spent on titration (mL); = 0.1 mmol/mL (NaOH);  msw  is the mass of a swollen sample (g).”

  1. Same commentaries can be observed as to the permeability equations in page 4 of the main manuscript, which differ to published literature in ion exchange membranes or membrane technology in general. Please revise.

Authors’ response:

This part is revised as well. The paragraph that defines the characteristics of P and P* is as follows (page 4):

“A two-compartment flow cell [62] was used for determining the integral diffusion permeability coefficient of membranes (P), which is usually defined as [56]:

(1)

where j is the flux density of an electrolyte measured when it diffuses through the membrane from the compartment containing a solution of concentration С into the compartment with initially distilled water; d is the membrane thickness. The confidence interval of the measurements is equal to 0.4ˣ10-8 cm2 s-1. The scheme of the cell, the details of the measurements and data processing are given in Supplementary materials.

P is a characteristic convenient for the practical use. However, in theoretical considerations, the differential (or local) diffusion permeability coefficient, P*, is often applied. The relation between these characteristics is given as  [56, 61], which leads to the following formula [61]:

(2)

In practice, for calculation P*, it is more convenient to use the relationship between P* and P in the form [60]:

,

(3)

with β=dlgP/dlgC, which is the slope of the lgP vs lgC dependence.”

  1. What is the functionality of the setup in Figure 1 of the main manuscript and the MK-40 CEM in comparison with the CEM being tested?

Authors’ response:

Thank you for this question. We revised the fragment describing the setup used for the measurements (page 4):

“The galvanodynamic current-voltage characteristics (CVC) of the membranes were obtained using a laboratory four-compartment flow-through cell [65]) shown in Figure 1. The current density was applied with the Autolab PGStat-100 (Metrohm Autolab B.V., Kanaalweg, The Netherlands) electrochemical complex, which was used also for recording the potential difference between the Ag/AgCl electrodes with the Luggin capillaries shown in Figure 1. The capillary tips were installed at a distance of about 0.8 mm from each side of the CEM under study (marked as CEM* in Figure 1). In addition to CVC, we have measured the change of pH of the feed solution, ΔpH, caused by its passage through the desalination channel (DC) of the cell. For this, pH measurements were carried out in two special flow-through cells equipped with a combination electrode; one of them was installed at the input, and the other at the output of the DC. Two auxiliary heterogeneous membranes (a MK-40 cation-exchange one and a MA-41 anion-exchange one, manufactured by Shchekinoazot JSC, Pervomaysky, Russia) were installed between the CEM under study. This allowed avoiding the transfer of the products of electrode reactions (the H+ and OH ions) from the electrode compartments to the compartments next to CEM*. Thus, the ΔpH value was due only to water splitting reactions occurred on the membranes forming the DC. The complete setup for electrochemical measurements is shown in Figure S4 of the Supplementary materials.”

  1. line 192, what do the authors mean by "tops and bottoms of relief"? Besides the misunderstanding of the expression, there is no distance of 40 microns in Figure2  being discussed. Please revise.

Authors’ response:

Indeed, the term “relief” is not suitable in this context. The sentence is changed to (line 202):

“The distance between the levels corresponding to the top of the “hills” and the bottom of the “valleys”, b, of such a surface is about 35 μm.”

  1. Please explain the images in Figure 3. Holes of about 150 microns are more than macropores (lines 199-200). These holes look more like a mechanical support of the commercial membrane than macropores or "structural defects" as commented later upon discussing transport performance. Please revise.

Authors’ response:

Sorry to give reason to think that the dark squares in Figs. 3 are macropores. In fact, they are ion-exchange material, and the long rectangles seen also in these figures are reinforcing threads. We revised the corresponding paragraph as follows (page 6):

“It appears that extended (lengthy) macropores are formed between the reinforcing threads and the ion-exchange material of the CJMC - 3 and CJMC – 5 membranes. These pores can be visualized by video recording the drying of swollen samples in air using the optical microscope described in section 2.3. Some of the frames of this video are presented in Figures 3a-f. In the case of swollen samples equilibrated with a 0.02 M sodium chloride solution (Figures 3a,d), these pores cannot be seen because they are filled with the solution, which is optically nontransparent in transmitted light. White bands appearing around the filaments of reinforcing fabric (Figure 3b,c, e,f) are caused by the air penetrating in the pores after water evaporation. The material of the filaments is seen as the dark regions, which are cleared by a layer of air between the filament and the meso-porous ion-exchange material. The latter remains dark, since it holds water well and air does not penetrate there. The images shown in Figures 3a-f allow concluding that drying of the swollen membrane begins at the intersections of the reinforcing filaments and then spreads along the filaments through the entire membrane. The pores between the filament and ion-exchange material were also detected on images of scanning microscopy of the heterogeneous MK-40 and MA-41 membrane [69].

Similar images are obtained in the case of CJMC – 3 membrane. However, in the case of the CMX membrane (Figure 3d-f), drying occurs fairly evenly over the surface and it is not possible to visualize the “rapid water evaporation” spots, although these membranes contain reinforcing fabric as well. Apparently, this is due to the fact that both the reinforcing cloth and the inert filler of the CMX ion-exchange composite material are made of the same PVC polymer, that is, they have good adhesion [44,70].”

  1. What do the continuous lines in Figure 5 and 6 represent?

Authors’ response:

These lines are drawn just to guide the eye. This addition is done in the captions to Figures 5 and 6 on pages 11 and 12, respectively.

  1. How are the electrolytes for the diffusion experiments selected?

Authors’ response:

Thank you for this question. The electrolytes are chosen because their ions are most often present in natural and wastewaters. This sentence is added in page 3.

  1. What purpose do the authors mean in line 357?

Authors’ response:

Thank you for your question. By «purpose» was meant the use of appropriate membranes for « electrodialysis of relatively dilute solutions». This clarification is added in page 12.

  1. Where was the pH measured? Bulk of near the surface? Please clarify.

Authors’ response:

pH measurement was carried out in two flow-through cells with a combination electrode; one of them was installed at the input and the other at the output of the desalination chamber. A schematic design of the set-up is shown in Figure S4 in Supplementary materials. This is explained now in page 4, please, see our response to your question about the functionality of the setup in Figure 1.

  1. English revision by a professional native translator is recommended.

Authors’ response:

Thank you, we tried to take into account and correct inaccuracies regarding the English language. We hope that they are now much smaller.

  1. State-of-the-art bibliography should be updated.

Authors’ response:

Your comment has been taken into consideration. Some of the references have been replaced by more recent ones.

Round 2

Reviewer 2 Report

The authors have answered previous reviewers' questions and the manuscript must thereby have been improved.

I still have the curiosity on the IEC measurement of AEM, or did the authors only measured the CEM? Can they comment a little bit?

Author Response

We thank the Reviewer for appreciating our efforts to improve the paper.

  1. I still have the curiosity on the IEC measurement of AEM, or did the authors only measured the CEM? Can they comment a little bit?

Authors’ response:

Thank you for this question. This work is devoted to the study of the characteristics of cation-exchange membranes. For this reason, the properties of auxiliary AEM (MA-41), which forms the desalination channel with the investigated CEM were not presented. However, we agree that these properties may be of interest for some readers. In the revised manuscript the paragraph describing characteristics of the MA-41 membrane is as follows (page 5):

“Although the paper is devoted to the study of CEMs, the properties of the MA-41 can be of interest. This membrane is produced by hot rolling of powdered AV-17 anion-exchange resin with low-pressure polyethylene used as an inert binder [44]. The reinforcing nylon fabric is made of filaments with the diameter of 30–50 μm. The main characteristics of the МА-41 membrane are as follows: the thickness of wet membrane is 450±50 μm; water content = 24 gН2О gdry-1, %; IEC = 1.22±0.6 mmol gsw-1; counterion transport number = 0.995 (at a 0.15 mol dm-3 NaCl solution).”
